# Cost and Effectiveness of Long-Term Care Following Integrated Discharge Planning: A Prospective Cohort Study

**DOI:** 10.3390/healthcare9111413

**Published:** 2021-10-21

**Authors:** Yu-Chun Wang, Wen-Ying Lee, Ming-Yueh Chou, Chih-Kuang Liang, Hsueh-Fen Chen, Shu-Chuan Jennifer Yeh, Chih-Liang Yaung, Kang-Ting Tsai, Joh-Jong Huang, Chi Wang, Yu-Te Lin, Shi-Jer Lou, Hon-Yi Shi

**Affiliations:** 1Center for Geriatrics and Gerontology, Kaohsiung Veterans General Hospital, Kaohsiung 81341, Taiwan; Trepakwang@gmail.com (Y.-C.W.); pitychou@gmail.com (M.-Y.C.); ck.vghks@gmail.com (C.-K.L.); ytlin@vghks.gov.tw (Y.-T.L.); 2Department of Healthcare Administration and Medical Informatics, Kaohsiung Medical University, Kaohsiung 80708, Taiwan; wyli@vghks.gov.tw (W.-Y.L.); chenhf@kmu.edu.tw (H.-F.C.); syehparis@gmail.com (S.-C.J.Y.); 3Department of Administration, Kaohsiung Veterans General Hospital, Kaohsiung 81341, Taiwan; 4Department of Geriatric Medicine, National Yang Ming Chiao Tung University School of Medicine, Taipei 11221, Taiwan; 5Aging and Health Research Center, National Yang Ming Chiao Tung University, Taipei 11221, Taiwan; 6Department of Business Management, National Sun Yat-sen University, Kaohsiung 80424, Taiwan; 7Department of Healthcare Administration, Asia University, Taichung 41354, Taiwan; clyaung@asia.edu.tw; 8Department of Geriatrics and Center for Integrative Medicine, Chi Mei Medical Center, Tainan 71004, Taiwan; irised57@gmail.com; 9Department of Health, Kaohsiung City Government, Kaohsiung 80251, Taiwan; jjhua@seed.net.tw; 10Department of Nursing, Kaohsiung Veterans General Hospital, Kaohsiung 81341, Taiwan; wchi@vghks.gov.tw; 11Graduate Institute of Technological and Vocational Education, National Pingtung University of Science and Technology, Pingtung 91201, Taiwan; 12Department of Medical Research, Kaohsiung Medical University Hospital, Kaohsiung 80708, Taiwan; 13Department of Medical Research, China Medical University Hospital, China Medical University, Taichung 40604, Taiwan

**Keywords:** long-term care, seamless transition, standard transition, medical costs, medical outcomes

## Abstract

Little is known about the effects of seamless hospital discharge planning on long-term care (LTC) costs and effectiveness. This study evaluates the cost and effectiveness of the recently implemented policy from hospital to LTC between patients discharged under seamless transition and standard transition. A total of 49 elderly patients in the standard transition cohort and 119 in the seamless transition cohort were recruited from November 2016 to February 2018. Data collected from medical records included the Multimorbidity Frailty Index, Activities of Daily Living Scale, and Malnutrition Universal Screening Tool during hospitalization. Multiple linear regression and Cox regression models were used to explore risk factors for medical resource utilization and medical outcomes. After adjustment for effective predictors, the seamless cohort had lower direct medical costs, a shorter length of stay, a higher survival rate, and a lower unplanned readmission rate compared to the standard cohort. However, only mean total direct medical costs during hospitalization and 6 months after discharge were significantly (*p* < 0.001) lower in the seamless cohort (USD 6192) compared to the standard cohort (USD 8361). Additionally, the annual per-patient economic burden in the seamless cohort approximated USD 2.9–3.3 billion. Analysis of the economic burden of disability in the elderly population in Taiwan indicates that seamless transition planning can save approximately USD 3 billion in annual healthcare costs. Implementing this policy would achieve continuous improvement in LTC quality and reduce the financial burden of healthcare on the Taiwanese government.

## 1. Introduction

Advances in medicine and sanitation in developed countries have achieved consistent annual increases in average life expectancy [1]. In 2013, the United States Medicare system implemented a payment coding system to improve the management of transitional care and to improve patient outcomes after discharge [2]. Five years later, a study by Bindman and Cox revealed that Medicare beneficiaries who had received transitional care within 31 to 60 days after hospital discharge had significantly lower adjusted total costs and mortality compared to those who had not received transitional care (*p* < 0.001) [3]. Naylor et al. examined the effectiveness of an advanced practice nurse-centered discharge planning and home follow-up intervention for elders at risk for hospital readmission; the authors reported that the intervention increased the time between discharge and readmission and decreased healthcare costs [4]. Forster et al. found that including a clinical nurse specialist in a medical team improved patient satisfaction but did not impact hospital efficiency or clinical outcomes [5]. Jack et al. found that a package of discharge services reduced hospital utilization within 30 days of discharge [6]. One literature review indicated that a structured discharge plan tailored to the individual patient is likely to decrease hospital stay, decrease readmission rate, and increase patient satisfaction but has an unknown impact on health outcomes [7]. Another systematic review of randomized, controlled, or quasi-experimental trials was performed in 2000–2009 to investigate how discharge planning from hospital to home affects health outcomes in patients aged 65 years or older [8]. In these patients, discharge planning had large effects on satisfaction but only moderate effects on quality of life and readmission rate.

Colemen and Boult proposed that transitional care should be based on a comprehensive plan for care and should include healthcare delivered by practitioners who are well trained in chronic care and have current information about the goals, preferences, and clinical status of the patient [9]. Transitional care should include logistical arrangements, the education of the patient and family, and coordination among the health professionals involved in the transition. To meet the needs of patients with complex care needs, transitional care planning should also consider caregiver needs. A discharge planning service that effectively utilizes and integrates medical resources is needed to provide an efficient pathway from acute care to LTC. Comprehensive discharge planning can achieve seamless, continuous care, which may include social services during transition, so that the patient and the family can leave the medical facility in a safe and timely manner and return home or transfer to another facility [6,7,8,9,10].

In Taiwan, a “Long-term Care Ten Years Plan 2.0 (LTC 2.0)” program was implemented in 2016 to provide additional LTC services, including the “Discharge Preparation Service Plan.” The LTC 2.0 program was designed for (1) people who have dementia and are older than 50 years; (2) indigenous people with functional limitations (aged 55 to 64 years); (3) people with disability (aged under 49 years); and (4) people with frailty (aged 65 years and over) [11]. In this “person-centered” LTC 2.0 program, care managers serve as gatekeepers of publicly funded care/support. After evaluating patient needs, care managers determine the appropriate benefit levels and care programs and then refer patients to relevant service resources. Once the services have been delivered, the care manager performs the standard reviews and monitors the care to ensure adequate service quality. Transition planning is performed by a team comprising a geriatrician, a nurse, a physical therapist, a social worker, and a nutritionist [12,13]. In the standard transition model, the elderly patient cannot request LTC until after a discharge to home. The care manager then assesses the need and makes arrangements for the LTC to provide the required services. On average, the wait for LTC services after discharge exceeds 4 weeks [14]. Since little is known about the impacts of a seamless discharge plan on LTC cost and effectiveness, however, this study purposed to evaluate the cost and effectiveness of the LTC policy recently implemented in Taiwan and compared the cost, effectiveness, and risk of poor outcomes between the standard transition model and the seamless transition model in elderly patients with disability after LTC discharge.

## 2. Materials and Methods

### 2.1. Study Design and Participants

This prospective cohort study was performed from November 2016 to February 2018 in southern Taiwan. The inclusion criteria were age 55 years or older, hospitalization for over 3 days, one or more impaired activities of daily living (ADL), and eligibility for the LTC 2.0 program. An additional inclusion criterion was agreement to participate in an LTC program after discharge to home. The exclusion criteria were (1) terminal stage of cancer; (2) critical condition or unconsciousness; (3) refusal to participate; (4) residence in LTC facility before hospitalization; (5) referral to LTC more than 1 week before discharge home; (6) care received from foreign caregiver after hospital discharge.

This study recruited 347 patients awaiting LTC service after hospital discharge. Of these, 191 patients were enrolled in the standard transition group, and 156 patients were simultaneously enrolled in the seamless transition group (Figure 1). In the standard transition group, six patients were lost to follow up, and 136 patients were unable to receive LTC service. Thus, 49 patients received the standard transition service. In the seamless transition group, 6 patients were lost to follow up, and 37 were unable to receive LTC service. Therefore, 119 patients received seamless transition service. The study protocol was approved by the institutional review board (VGHKS18-EM6-01), and informed consent was obtained from each participant before enrollment in the study.

### 2.2. Study Measures

#### 2.2.1. Multimorbidity Frailty Index (MFI)

Frailty is a common geriatric syndrome associated with increased risk of catastrophic declines in health and function in older adults. Frailty syndrome occurs when cumulative negative factors exceed cumulative positive factors. To construct the MFI, this study used the cumulative deficit model, which is among the most common models of frailty [15,16]. The MFI defines frailty according to disease state as well as disability signs and symptoms. Based on the presence of each deficit as a proportion of the total, the MFI is calculated as a continuous variable ranging from 0 to 1; a high value indicates high frailty.

#### 2.2.2. Activities of Daily Living (ADL)

The ADL scale used in the Taiwan LTC information system is a modified version of the Katz ADL scale. The scale assesses six primary and psychosocial functions: bathing, dressing, going to toilet, transferring, feeding, and continence. All six ADL functions are quantified using an evaluative form [17,18,19,20]. Scores for the Taiwan version of the ADL scale range from 0 to 6, with a higher score indicating higher severity of ADL impairment. Patients were defined as completely independent, mildly disabled, moderately disabled, or severely disabled if they required no assistance, assistance with 1–2 activities, assistance with 3–4 activities, or assistance with 5–6 activities, respectively.

#### 2.2.3. Malnutrition Universal Screening Tool (MUST)

The MUST was used to assess malnutrition risk [21]. The MUST score is based on three parameters: body mass index (BMI) at presentation, percentage of total body weight loss in the past 3–6 months, and presence of acute disease in the past 5 days [22]. Scores of 0, 1, and ≥2 indicate low, moderate, and high risk of malnutrition, respectively. In our study, patients with a MUST score ≥1 were considered malnourished [23]. The MUST has been widely used in medical research [22,23,24].

#### 2.2.4. Medical Record Review

Medical records were reviewed to collect demographic and clinical data. Demographic data collection included age, gender, BMI, education, marital status, family support, smoking, and drinking. Clinical data collection included Charlson Comorbidity Index (CCI) score, discharge with urinary catheter, medical resource utilization, medical outcome, and comorbidities. Medical resource utilization data collection included lengths of stay (LOS) before discharge, total medical direct costs during hospitalization, total outpatient costs within 6 months after discharge, total inpatient costs within 6 months after discharge, total emergency room costs within 6 months after discharge, and total medical direct costs before discharge and within 6 months after discharge. Data for medical outcome included readmission 14 days after discharge, readmission 30 days after discharge, readmission 90 days after discharge, readmission 180 days after discharge, and mortality. Comorbidities were determined according to primary and secondary ICD-9-CM diagnostic codes, excluding those related to cancer. Diagnostic codes were then used to calculate CCI score as modified by Deyo et al. [25].

### 2.3. Statistical Analysis

The unit of statistical analysis in this Taiwanese study was the individual elderly patient [26,27]. Sample size calculations determined that, for a power of 0.80, a minimum sample of 120 study participants was needed with an alpha of 0.05 [28]. Descriptive statistical analysis included demographic characteristics, clinical characteristics, medical resource utilization, and medical outcomes. The standard administrative claims data required by the Taiwan Bureau of National Health Insurance include fees for the following: physician, radiology, physical therapy, hospital room, pharmacy, laboratory, special materials, and others. Total direct medical costs during hospitalization and within 6 months after discharge included total direct medical costs during hospitalization, total outpatient costs within 6 months after discharge, total inpatient costs within 6 months after discharge, and total emergency room costs within 6 months after discharge. To reflect changes in real dollar value, all dollar values were converted to their equivalent 2020 values; New Taiwan Dollar values were then converted to USD values at the average exchange rate over the 3-year period of 2016–2018 (TWD 30.5 = USD 1).

Length of hospital stay and total direct medical costs were included in simultaneous analyses of their associations with patient characteristics. When norms were set for length of hospital stay and total direct medical costs, data for normal controls were depicted as a natural logarithmic transformation because the data were highly skewed and did not follow a normal distribution. Thereafter, all data were converted back to natural numbers for convenient analysis. Multiple logistic regression and Cox proportional hazards model were also employed to conduct the significant predictors of medical resource utilization and medical outcomes after adjustment of effective predictors. Additionally, an incidence-based approach was used to compare the per-patient economic burden of total direct medical costs between the seamless transition cohort and the standard transition cohort during 1 year after rehabilitation. Statistical analyses were performed with SPSS version 23.0 (SPSS Inc., Chicago, IL, USA). All tests were two-sided, and *p* values less than 0.05 were considered statistically significant.

## 3. Results

Table 1 shows that demographic characteristics and clinical characteristics did not significantly differ between the seamless transition cohort and the standard transition cohort. Mean age was 77.77 years (standard deviation, SD 11.32 years) in the seamless cohort and 80.88 years (SD 9.96 years) in the standard cohort. During the study period, the seamless cohort had 70 males (58.8%) and the standard cohort had 33 males (67.3%). In the standard cohort, study characteristics did not significantly differ between the 49 patients with follow up and the 135 patients without follow up. Similarly, in the seamless cohort, study characteristics did not significantly differ between the 119 patients with follow up and the 37 without follow up (Appendix A).

Furthermore, Table 2 and Table 3 show that, after adjustment for effective predictors, the seamless cohort had lower direct medical costs, a shorter LOS, a higher survival rate, and a lower unplanned readmission rate compared to the standard cohort. However, the only difference that reached statistical significance was total direct medical costs during hospitalization and during the 6-month follow-up period after discharge (both *p* = 0.041). 

Additionally, Table 4 further shows that mean total direct medical costs during hospitalization and 6 months after discharge were significantly (*p* < 0.001) lower in the seamless cohort (USD 6192) compared to the standard cohort (USD 8361). The per-patient annual economic burden of total direct medical costs of LTC delivered to disabled elderly patients after integrated discharge planning ranges from approximately USD 2.9 billion to USD 3.3 billion.

## 4. Discussion

The two cohorts did not significantly differ in demographic or clinical characteristics when selecting study samples representative of the overall population. Furthermore, in both the seamless cohort and the standard cohort, study characteristics did not significantly differ between those with and without follow up. To our knowledge, this study is the first to prospectively investigate LTC cost and effectiveness after integrated discharge planning. This study revealed that the seamless cohort had significantly (*p* < 0.05) lower total direct medical costs compared to the standard cohort. However, the two cohorts did not significantly differ in medical resource utilization or medical outcomes during the study period.

Naylor et al. demonstrated that successful transitional care interventions (e.g., assigning a nurse as the clinical manager or leader of care and including in-person home visits to discharged patients) can reduce readmissions 30 or more days after discharge [29]. Verhaegh et al. suggested that, to reduce short-term readmissions, transitional care should consist of high-intensity interventions that include care coordination by a nurse, communication between the primary care provider and the hospital, and a home visit within 3 days after discharge [30]. Low et al. reported that patients enrolled in a transitional home care program within the first 3 months after discharge had significantly (*p* < 0.001) lower hospital admissions and emergency department admissions compared to patients discharged without enrollment in such a program [31]. Our findings add to growing evidence that multi-disciplinary transitional care programs reduce medical resource utilization [29,30,31]. A multidisciplinary approach based on family medicine and geriatric medicine paradigms may have contributed to the effectiveness of this transitional care program.

Although previous studies have shown that appropriate transitional care after hospital discharge can decrease readmission rates, decrease LOS in patients with chronic diseases, and increase satisfaction in patients and medical staff [2,29,30,31,32], studies of the role of transitional care in medical outcomes of geriatric populations have reported inconsistent results. The seamless transition cohort and the standard transition cohort in the present study did not significantly differ in hospital LOS, in survival rate, or in unplanned hospital readmission within 14, 30, 90, or 180 days after discharge. Braet et al. systemically reviewed 47 randomly selected studies of the effect of implementing hospital discharge plans [33]. They found that interventions designed to improve care during the transition from hospital to home significantly reduced hospital readmission risk (relative risk, RR = 0.77, *p* < 0.001) but did not significantly affect emergency department visits or mortality after discharge. They also suggested that these interventions should be implemented during hospitalization and continued after discharge. A recent systematic study performed across 13 databases retrieved published and unpublished studies of discharge planning published in English during 2000–2015 [34]. The study indicated that the planning of home nursing care for older patients discharged home increases hospital LOS but does not decrease the readmission rate or improve quality of life. Zurlo and Zuliani suggested that, to reduce the risk of negative outcomes after discharge, a hospital organization dedicated to achieving the optimal discharge conditions for frail older patients must (1) adequately and comprehensively assess clinical/social/care conditions; (2) consider the expectations of patients and their relatives; (3) formalize the roles of the institution and the roles of care teams responsible for planning and coordinating discharge; (4) acquire adequate knowledge of programs for managing transitional care; and (5) communicate with and provide information to patients transitioning between different care settings, including hospital, home, and community [35].

Although potential confounders of transitional care interventions vary considerably across different systems of care, the findings of the present study indicate the valuable role of a transitional care program in reducing medical resource utilization by elderly patients. Disease progression varies widely among elderly patients. Some complications that trigger the worsening of functional status and quality of life are preventable, and failure to eliminate preventable complications can increase the negative effects of a disease at the time of readmission, including disease severity, morbidity, and mortality [36,37,38]. That is, some complications substantially increase healthcare resource consumption. Thus, discharge planning and management to improve care continuity has an important role in eliminating triggers of disease progression. For example, a transitional care program for stroke patients reportedly reduced 30-day readmissions by 48% [39]. Our findings contribute to the literature by suggesting that seamless transitional care is an important tool for reducing medical resource utilization for elderly patients.

Although some factors investigated in this study did not reveal statistically significant associations with unplanned readmissions after discharge, with mortality during hospitalization, or with mortality within 6 months after discharge, our results did confirm that the seamless transition cohort had significantly (*p* < 0.05) lower total direct medical costs during the study period compared to the standard transition cohort. A possible explanation is that the seamless cohort may have had relatively higher awareness of care required for patients with their disability. The seamless transition cohort received LTC immediately after hospital discharge and a follow-up evaluation 1 month after discharge. Geriatric conditions were highly prevalent and associated with poor health outcomes after discharge. Early recognition of these conditions in older patients hospitalized for acute care can improve the transition to the general practitioner, improve health outcomes, and reduce the healthcare burden of readmitted older patients [40].

In comparisons of the two models of discharge planning to facilitate the transition to LTC, the average total direct medical cost per patient was USD 2168.50 lower in the seamless transition group compared to the standard transition group. The total direct medical costs for 1 year were USD 4337.0 lower in the seamless transition group compared to the standard transition group. According to annual statistical data reported by the Ministry of Health and Welfare, 686,352 to 765,218 people were eligible for Long-Term Care 2.0 in 2018. If every eligible patient received seamless transition care, the economic burden of healthcare would decrease by an estimated USD 2.9 billion to USD 3.3 billion annually.

Despite the importance of our findings, this study had some limitations. First, this was a prospective cohort study performed in cooperation with the government promotion of Long-Term Care 2.0 in recent years. The objective of this collaborative effort was to improve the continuity and quality of healthcare after hospital discharge. The LTC 2.0 program is expected to achieve this objective by enabling patients to complete relevant evaluations for care before discharge. However, recruiting patients who met the criteria for enrolment in the standard transition cohort after discharge was complicated by timing conflicts, e.g., conflicts between the timing for implementing the program and the timing for obtaining signatures for the consent form. Another limitation is the small sample size, which restricted the extent to which the findings can be generalized to larger populations. Future studies are needed to examine outcomes, patient characteristics, clinical characteristics, quality of care, and related factors in a larger population.

## 5. Conclusions

This prospective cohort study was the first to investigate the impact of a program for the seamless transition of elderly patients to LTC after hospital discharge in a Taiwanese population. Compared to standard transitional care, seamless transitional care can be a more effective approach for reducing total medical costs for elderly patients. However, evidence of the benefits of seamless transitional care on the length of stay, readmission, and mortality in elderly patients remains insufficient. Since elderly patients exhibit individual variation in disease conditions, hospital readmission risk, and mortality risk, tailoring transitional care interventions to specific disease-related factors in each patient is essential for effective seamless transitional care. Based on the findings of this study, we recommend a nationwide expansion of seamless transitional care, which would not only increase the quality of care delivered under the Long-Term Care 2.0 program, but would also reduce the healthcare cost burden in the general population.

## Figures and Tables

**Figure 1 healthcare-09-01413-f001:**
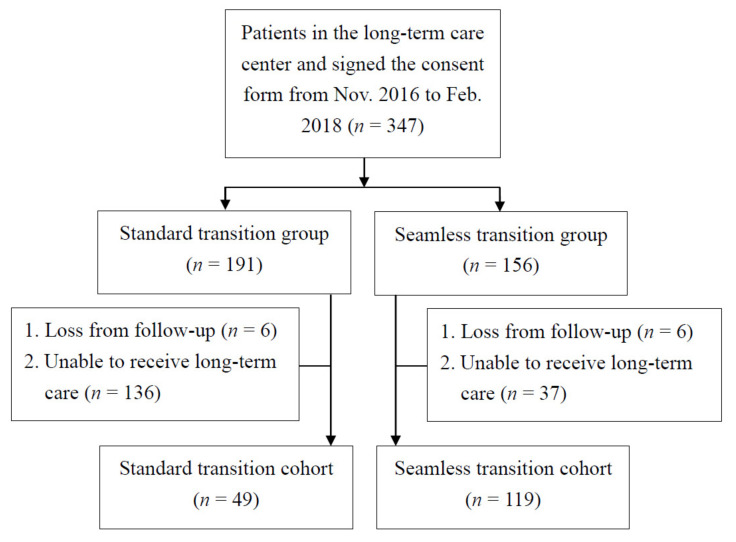
Flowchart of sample selection procedure.

**Table 1 healthcare-09-01413-t001:** Comparison of patient characteristics between standard transition cohort and seamless transition cohort (*n* = 168).

Variables		Standard Transition Cohort (*n* = 49)	Seamless Transition Cohort (*n* = 119)	*p* Value
**Demographic characteristics**				
Age, years		80.88 ± 9.96	77.77 ± 11.32	0.097
Gender	Male	33(67.3%)	70(58.8%)	0.392
	Female	16(32.7%)	49(41.2%)	
Body mass index, km/m^2^		22.31 ± 4.24	22.79 ± 3.97	0.486
Education, years		7.88 ± 5.37	6.50 ± 4.9	0.110
Marital status	Single	5(10.2%)	4(3.4%)	0.208
	Married	32(65.3%)	79(66.4%)	
	Widowed	12(24.5%)	36(30.3%)	
Family support *		1.08 ± 0.53	1.08 ± 0.51	0.979
Smoking	Yes	3(6.1%)	17(14.3%)	0.221
Drinking	Yes	5(10.2%)	7(5.9%)	0.338
**Clinical characteristics**				
Charlson Comorbidity Index, score		4.25 ± 2.64	4.60 ± 3.28	0.506
Multimorbidity Frailty Index		0.15 ± 0.87	0.14 ± 0.11	0.849
Activities of Daily Living *	0	3(6.1%)	15(12.6%)	0.059
	1	8(16.3%)	24(20.2%)	
	2	20(40.8%)	25(21.0%)	
	3	18(36.7%)	55(46.2%)	
Malnutrition Universal Screening Tool *	0	33(67.3%)	78(65.5%)	0.265
	1	4(8.2%)	20(16.8%)	
	2	12(24.5%)	21(17.6%)	
Discharged with urinary catheter	Yes	10(20.4%)	34(28.6%)	0.368
**Medical resource utilization**				
Length of stay before discharge, days		25.71 ± 23.40	22.61 ± 15.90	0.399
Total direct medical costs during hospitalization, USD		148,231.78 ± 105,067.45	135,976.91 ± 127,758.22	0.554
Total outpatient costs at 6 months after discharge, USD		21,237.92 ± 13,778.73	15,316.95 ± 14,330.01	0.015
Total inpatient costs at 6 months after discharge, USD		76,329.61 ± 209,050.34	33,066.32 ± 60,084.40	0.040
Total emergency room costs at 6 months after discharge, USD		9215.75 ± 12,423.25	4515.60 ± 6687.79	0.002
Total medical direct costs before and 6 months after discharge, USD		255,015.05 ± 267,362.66	188,875.78 ± 152,821.57	0.045
**Medical outcomes**				
Readmission within 14 days after discharge	Yes	2(4.1%)	13(10.9%)	0.235
Readmission within 30 days after discharge	Yes	8(16.3%)	23(19.3%)	0.813
Readmission within 90 days after discharge	Yes	11(22.4%)	37(31.1%)	0.348
Readmission within 180 days after discharge	Yes	21(42.9%)	46(38.7%)	0.740
Mortality	Yes	5(10.2%)	18(15.1%)	0.551

* Family support: living or staying with spouse (yes 1, no 0), living with parents (yes 1, no 0), children (yes 1, no 0); Activities of Daily Living: 0 completely independent, 1 mild disability, 2 moderate disability, 3 severe disability; Malnutrition Universal Screening Tool: 0 low risk, 1 mild risk, 2 high risk.

**Table 2 healthcare-09-01413-t002:** Multivariate analysis of medical resource utilization and mortality in the two cohorts after adjustment for patient study characteristics.

	Unplanned Readmission within 14 Days			Unplanned Readmission within 30 Days			Unplanned Readmission within 90 Days			Unplanned Readmission within 180 Days		
Variable	OR	95% C.I.	*p* Value	OR	95% C.I.	*p* Value	OR	95% C.I.	*p* Value	OR	95% C.I.	*p* Value
Transition cohort, seamless vs. standard	3.53	0.69–18.00	0.130	1.19	0.40–3.57	0.759	2.03	0.82–5.01	0.125	0.96	0.46–1.98	0.906
Age	1.01	0.96–1.07	0.639	1.03	0.99–1.07	0.217	1.02	0.99–1.06	0.168	1.03	1.01–1.06	0.032
Gender, male vs. female	0.08	0.01–0.54	0.010	0.39	0.13–1.18	0.094	0.96	0.42–2.19	0.929	1.01	0.52–1.94	0.986
Body mass index, km/m^2^	1.01	0.85–1.18	0.991	1.02	0.92–1.14	0.664	0.96	0.89–1.03	0.250	0.94	0.87–1.01	0.055
Education, years	1.02	0.90–1.16	0.729	1.03	0.94–1.14	0.501	1.02	0.95–1.10	0.624			
Marital status												
Married vs. single				0.12	0.02–0.91	0.040	0.12	0.02–0.78	0.026	0.27	0.05–1.48	0.130
Widowed vs. single				0.07	0.01–0.62	0.017	0.10	0.01–0.66	0.017	0.19	0.03–1.12	0.066
Family support *	0.40	0.12–1.36	0.143	0.29	0.12–0.72	0.008	0.50	0.24–1.04	0.064	0.71	0.37–1.35	0.294
Smoking, yes vs. no	0.31	0.03–2.89	0.302	0.27	0.05–1.55	0.142	0.29	0.07–1.20	0.088			
Drinking, yes vs. no	0.48	0.04–5.46	0.553	0.82	0.13–5.12	0.834	0.62	0.11–3.51	0.592			
Charlson Comorbidity Index, score	1.14	0.95–1.36	0.168	1.16	1.01–1.35	0.046	1.12	0.99–1.27	0.079			
Multimorbidity Frailty Index	0.13	0.01–87.00	0.541	0.58	0.01–11.20	0.840	10.60	0.16–70.94	0.270			
Activities of Daily Living *												
1 vs. 0	0.37	0.05–2.99	0.350	0.27	0.05–1.63	0.153	0.75	0.19–2.96	0.681	2.20	6.31–7.64	0.217
2 vs. 0	0.48	0.07–3.47	0.466	0.24	0.04–1.31	0.098	0.69	0.18–2.65	0.585	1.20	0.35–4.12	0.772
3 vs. 0	0.22	0.03–1.41	0.109	0.69	0.16–2.94	0.615	0.99	0.30–3.25	0.992	2.14	0.70–6.59	0.183
Malnutrition Universal Screening Tool *												
1 vs. 0	0.91	0.14–6.21	0.926	2.24	0.55–9.13	0.259						
2 vs. 0	0.22	0.02–2.83	0.247	0.60	0.13–2.74	0.509						
Discharged with urinary catheter												
Yes vs. no	2.39	0.68–8.39	0.175	3.18	1.20–8.43	0.020						

HR, hazard ratio; CI, confidence interval; * Family support: living or staying with spouse (yes 1, no 0), parents (yes 1, no 0), and children (yes 1, no 0); Activities of Daily Living: 0 completely independent, 1 mild disability, 2 moderate disability, 3 severe disability; Malnutrition Universal Screening Tool: 0 low risk, 1 mild risk, 2 high risk.

**Table 3 healthcare-09-01413-t003:** Multivariate analysis of unplanned readmission after discharge: comparison of the two cohorts after adjustment for patient study characteristics.

	Unplanned Readmission within 14 Days			Unplanned Readmission within 30 Days			Unplanned Readmission within 90 Days			Unplanned Readmission within 180 Days		
Variable	OR	95% C.I.	*p* Value	OR	95% C.I.	*p* Value	OR	95% C.I.	*p* Value	OR	95% C.I.	*p* Value
Transition cohort, seamless vs. standard	3.53	0.69–18.00	0.130	1.19	0.40–3.57	0.759	2.03	0.82–5.01	0.125	0.96	0.46–1.98	0.906
Age	1.01	0.96–1.07	0.639	1.03	0.99–1.07	0.217	1.02	0.99–1.06	0.168	1.03	1.01–1.06	0.032
Gender, male vs. female	0.08	0.01–0.54	0.010	0.39	0.13–1.18	0.094	0.96	0.42–2.19	0.929	1.01	0.52–1.94	0.986
Body mass index, km/m^2^	1.01	0.85–1.18	0.991	1.02	0.92–1.14	0.664	0.96	0.89–1.03	0.250	0.94	0.87–1.01	0.055
Education, years	1.02	0.90–1.16	0.729	1.03	0.94–1.14	0.501	1.02	0.95–1.10	0.624			
Marital status												
Married vs. single				0.12	0.02–0.91	0.040	0.12	0.02–0.78	0.026	0.27	0.05–1.48	0.130
Widowed vs. single				0.07	0.01–0.62	0.017	0.10	0.01–0.66	0.017	0.19	0.03–1.12	0.066
Family support *	0.40	0.12–1.36	0.143	0.29	0.12–0.72	0.008	0.50	0.24–1.04	0.064	0.71	0.37–1.35	0.294
Smoking, yes vs. no	0.31	0.03–2.89	0.302	0.27	0.05–1.55	0.142	0.29	0.07–1.20	0.088			
Drinking, yes vs. no	0.48	0.04–5.46	0.553	0.82	0.13–5.12	0.834	0.62	0.11–3.51	0.592			
Charlson Comorbidity Index, score	1.14	0.95–1.36	0.168	1.16	1.01–1.35	0.046	1.12	0.99–1.27	0.079			
Multimorbidity Frailty Index	0.13	0.01–87.00	0.541	0.58	0.01–11.20	0.840	10.60	0.16–70.94	0.270			
Activities of Daily Living *												
1 vs. 0	0.37	0.05–2.99	0.350	0.27	0.05–1.63	0.153	0.75	0.19–2.96	0.681	2.20	6.31–7.64	0.217
2 vs. 0	0.48	0.07–3.47	0.466	0.24	0.04–1.31	0.098	0.69	0.18–2.65	0.585	1.20	0.35–4.12	0.772
3 vs. 0	0.22	0.03–1.41	0.109	0.69	0.16–2.94	0.615	0.99	0.30–3.25	0.992	2.14	0.70–6.59	0.183
Malnutrition Universal Screening Tool *												
1 vs. 0	0.91	0.14–6.21	0.926	2.24	0.55–9.13	0.259						
2 vs. 0	0.22	0.02–2.83	0.247	0.60	0.13–2.74	0.509						
Discharged with urinary catheter												
Yes vs. no	2.39	0.68–8.39	0.175	3.18	1.20–8.43	0.020						

OR, odds ratio; CI, confidence interval * Family support: living or staying with spouse (yes 1, no 0), parents (yes 1, no 0), children (yes 1, no 0); Activities of Daily Living: 0 completely independent, 1 mild disability, 2 moderate disability, 3 severe disability; Malnutrition Universal Screening Tool: 0 low risk, 1 mild risk, 2 high risk.

**Table 4 healthcare-09-01413-t004:** Economic burdens and difference in various medical costs during hospitalization and after discharge in seamless transition cohort and standard transition cohort (*n* = 168).

Mean Value	Seamless Transition Cohort (*n* = 119)	Standard Transition Cohort (*n* = 49)	Difference * (Seamless–Standard)
Total direct medical costs during hospitalization	4458.26	4860.06	−401.80
Total outpatient costs at 6 months after discharge	502.20	696.32	−194.12
Total inpatient costs at 6 months after discharge	1084.14	2502.61	−1418.47
Total emergency costs at 6 months after discharge	148.05	302.15	−154.10
Total direct medical costs at 6 months after discharge	6192.65	8361.15	−2168.50

Economic burden: USD—2168.50 * 2/patient/year * (686,352~765,218 risk population/year) = USD—2,976,708,624~USD—3,318,750,466/year; * All *p* values < 0.001.

## Data Availability

Data related to the study are available from the authors upon reasonable request.

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
