# Peer review of "Cost and Effectiveness of Long-Term Care Following Integrated Discharge Planning: A Prospective Cohort Study"

_healthcare, 2021, doi:10.3390/healthcare9111413_

Round 1

Reviewer 1 Report

1.As a general comment, there are minor mistakes in written and grammar. It is better for the author to review the whole article carefully and find the errors or mistakes in grammar or ask someone whose native language is English to correct the errors.

2.Because there is little known about the impact of a seamless discharge plan on the cost and effectiveness of LTC,it is significant for this study to evaluate the cost and effectiveness of the newly proposed LTC policy and to compare risk of these outcomes between the standard transition model and the seamless transition model in elderly patients with disability after LTC discharge. The result suggest that this policy would achieve continuous improvement in long-term care (LTC) quality and reduce the financial burden of healthcare on the Taiwan government.

3.The author must describe that the sample size this cohort study needs and how to calculate it.

Author Response

1. As a general comment, there are minor mistakes in written and grammar. It is better for the author to review the whole article carefully and find the errors or mistakes in grammar or ask someone whose native language is English to correct the errors.

Ans:

On behalf of our co-authors, I would like to thank you for your excellent suggestions for improving our manuscript. As advised, the resubmitted manuscript has been reviewed and edited by a professional technical editor who is a native English speaker. Thank you.

2. Because there is little known about the impact of a seamless discharge plan on the cost and effectiveness of LTC, it is significant for this study to evaluate the cost and effectiveness of the newly proposed LTC policy and to compare risk of these outcomes between the standard transition model and the seamless transition model in elderly patients with disability after LTC discharge. The results suggest that this policy would achieve continuous improvement in long-term care (LTC) quality and reduce the financial burden of healthcare on the Taiwan government.

Ans:                      

Thank you for your comment. The Introduction section of the revised manuscript summarizes the study motivation and findings as observed by the reviewer (lines 7-11, page 6).

3. The author must describe that the sample size this cohort study needs and how to calculate it.

Ans:

Our sample size calculations revealed that, for a power of 0.80 and an alpha value of 0.05, at least 120 study participants were needed. This statement in the Section of Statistical Analysis (lines 18-19, page 9) has been revised accordingly. Thank you.

Reviewer 2 Report

The research is a cohort study evaluating the effectiveness and costs of different care programmes for the elderly. The methodological framework is robust, the analyses carried out are correct and the initial assumptions are confirmed in the results, which are of some interest as they point to the possibility of saving public money. The only limitation of the study, indicated by the authors themselves, is the relatively small number of participants, but conducting a study of this type involves high costs and a high percentage of subjects leaving the study during its course.

Author Response

The research is a cohort study evaluating the effectiveness and costs of different care programmes for the elderly. The methodological framework is robust, the analyses carried out are correct and the initial assumptions are confirmed in the results, which are of some interest as they point to the possibility of saving public money. The only limitation of the study, indicated by the authors themselves, is the relatively small number of participants, but conducting a study of this type involves high costs and a high percentage of subjects leaving the study during its course.

Ans:

Thank you for your encouraging comments.

Reviewer 3 Report

I would like to thank the authors for an interesting manuscript. I have the following comments: 

  1. Introduction is unfocused. It would most probably benefit from first presenting the synthesis of other research on discharge planning and then present the LTC 2.0 immediately before the study aim. 
  2. The introduction does not provide an enough rationale for inclusion of the outcomes selected. Please develop the introduction in this respect.
  3. Exclusion criteria #6 - what is meant by this and why is it important? Relates to point 2 above. 
  4. The description of the ADL instrument is confusing. The scale used is the Katz ADL index as concerns items included (refs 18 and 19). However, ref 20 and the rating scale concerns InterRAI,  which something else. The authors need to clarify how and why the Katz ADL scale was used without its original rating scale. They also need to clarify how the reliability of the new rating scale was tested. Also, Katz ADL scale is constructed as a hierarchical scale, with the activities learned last in the end of the scale (i.e. bathing) and the ones learned first in the beginning (i.e. feeding) in the beginning. Dependence is most often developed in the reverse order, with some deviations. In this context it is not self evident to use the categorization "mildly disabled" etc., based on number of activities the person is dependent in. There are many studies out there describing the scale and the authors need to sort these issues out (e.g. https://journals.sagepub.com/doi/abs/10.1177/0308022619853513 or  BMC Public Health. 2019 Nov 4;19(1):1446. doi: 10.1186/s12889-019-7815-9 or https://pubmed.ncbi.nlm.nih.gov/17366073/
  5. Statistical analysis: The authors state that the unit of analysis was the individual patient. However, all analyses was conducted on group level. No references support this statement and its relation to the analyses performed. Please clarify and provide references to support.
  6. Referring to #4 above on the ADL scale, table 1 is difficult to understand since the ADL score is presented as 0-3. What does this stand for here? The authors need to clarify this under the description of the scale and scores. 
  7. Table 2-3: Figures after adjustment of effective predictors are presented. The adjustment is not mentioned in the statistical analysis section. This is required. 
  8. Discussion: Substantial parts of the discussion repeats or extends the literature review rather than discussing the results. This part of the manuscript requires substantial revision throughout to ensure that the results are in focus
  9. Discussion: A discussion about the differences in group sizes is required. 
  10. Conclusion: repeats the findings but provides very little conclusion. Recommendations made are not supported by the study findings.  

Author Response

I would like to thank the authors for an interesting manuscript. I have the following comments:

1. Introduction is unfocused. It would most probably benefit from first presenting the synthesis of other research on discharge planning and then present the LTC 2.0 immediately before the study aim.

Ans:

As advised by the reviewer, the Introduction of the revised manuscript briefly synthesizes the recent literature on discharge planning and introduces the seamless transition model before stating the aim of the study. Thank you for your suggestions.

2. The introduction does not provide an enough rationale for inclusion of the outcomes selected. Please develop the introduction in this respect.

Ans:

As advised, the Introduction section of the revised manuscript includes an expanded discussion of our rationale for the selection of outcomes measures in this study (lines 3-8, page 4). Thank you.

3. Exclusion criteria #6 - what is meant by this and why is it important? Relates to point 2 above.

Ans:

Text related to criterion #6 has been deleted.

4. The description of the ADL instrument is confusing. The scale used is the Katz ADL index as concerns items included (refs 18 and 19). However, ref 20 and the rating scale concerns InterRAI, which something else. The authors need to clarify how and why the Katz ADL scale was used without its original rating scale. They also need to clarify how the reliability of the new rating scale was tested. Also, Katz ADL scale is constructed as a hierarchical scale, with the activities learned last in the end of the scale (i.e. bathing) and the ones learned first in the beginning (i.e. feeding) in the beginning. Dependence is most often developed in the reverse order, with some deviations. In this context it is not self evident to use the categorization "mildly disabled" etc., based on number of activities the person is dependent in. There are many studies out there describing the scale and the authors need to sort these issues out (e.g. https://journals.sagepub.com/doi/abs/10.1177/0308022619853513 or BMC Public Health. 2019 Nov 4;19(1):1446. doi: 10.1186/s12889-019-7815-9 or https://pubmed.ncbi.nlm.nih.gov/17366073/

Ans:                                                                                                             

The ADL data source was the long-term care information system officially used in Taiwan. The system uses the Katz ADL scale for scoring and then converts the scores to the equivalent Barthel Index (ref. #20 has been deleted). Thus, the modified ADL scale has four categories. Although the implications of this covariate may not be self-evident, we did not adjust this covariate to ensure a faithful representation of the original system data. Other reviewer comments are also addressed in lines 4-11, page 8. Additionally, we thank the reviewer for the literature references, and we have cited them in the revised manuscript.

5. Statistical analysis: The authors state that the unit of analysis was the individual patient. However, all analyses was conducted on group level. No references support this statement and its relation to the analyses performed. Please clarify and provide references to support.

Ans:

Thank you for your advice. Previous studies in which the unit of statistical analysis was the individual patient include Lou SJ, et al. (2020) and Kuo YH, et al. (2021).  Our inclusion criteria were age 55 years or older, hospitalization for over 3 days, one or more impaired activities of daily living (ADL), and eligibility for the long-term care (LTC) 2.0 program. The references in the revised manuscript have been revised accordingly (references # 26, 27).

6. Referring to #4 above on the ADL scale, table 1 is difficult to understand since the ADL score is presented as 0-3. What does this stand for here? The authors need to clarify this under the description of the scale and scores.

Ans:

Thank you for noting this typo. Classify as four descriptions: Patients were classified as 0 completely independent, 1 mildly disabled, 2 moderately disabled, or 4 severely disabled if they required no assistance, assistance with 1-2 activities, assistance with 3-4 activities, or assistance with 5-6 activities, respectively. The revised paragraph clarifies the definitions in Table 1.

7. Table 2-3: Figures after adjustment of effective predictors are presented. The adjustment is not mentioned in the statistical analysis section. This is required.

Ans:

As advised, the revised section 2.3 Statistical Analysis clarifies that the figures in Tables 2-3 were adjusted for effective predictors (lines 17-18, page 10). Thank you.

8. Discussion: Substantial parts of the discussion repeats or extends the literature review rather than discussing the results. This part of the manuscript requires substantial revision throughout to ensure that the results are in focus.

Ans:

As advised by the reviewer, the Discussion section in the revised manuscript provides a more detailed and focused discussion of the results of this study (lines 13-20, page 14 & lines 1-4, page 15). Thank you.

9. Discussion: A discussion about the differences in group sizes is required.

Ans:

The size differences between the two groups are discussed in lines 10-14, page 16 of the revised manuscript. Thank you for your comment.

10. Conclusion: repeats the findings but provides very little conclusion.

Recommendations made are not supported by the study findings.

Ans:

To address this comment by Reviewer #3, the discussion of the practical implications of this study in the Conclusions section has been expanded and clarified, and specific recommendations are given for improving seamless discharge from LTC.